# Identification and management of young infants with possible serious bacterial infection where referral was not feasible in rural Lucknow district of Uttar Pradesh, India: An implementation research

Shally Awasthi[1]*, Naveen Kesarwani[1], Raj Kumar Verma[1], Girdhar Gopal Agarwal[2], Luxmi Shanker Tewari[1], Ravi Krishna Mishra[1], Lalji Shukla[1], Arun Kumar Raut[1], Shamim Ahmad Qazi[3], Samira Aboubaker[3], Yasir Bin Nisar[4], Rajiv Bahl[4], Monika Agarwal[5]

1 Department of Pediatrics, King George's Medical University, Lucknow, Uttar Pradesh, India, 2 Department of Statistics, University of Lucknow, Lucknow, Uttar Pradesh, India, 3 Department of Maternal Newborn Child and Adolescent Health, World Health Organization, Geneva, Switzerland, 4 Department of Maternal Newborn Child and Adolescent Health and Aging, World Health Organization, Geneva, Switzerland, 5 Department of Social and Preventive Medicine, King George's Medical University, Lucknow, Uttar Pradesh, India

☯ These authors contributed equally to this work.
* shally07@gmail.com

## Abstract

### Background

Based on World Health Organization guidelines, Government of India recommended management of possible serious bacterial infection (PSBI) in young infants up to two months of age on an outpatient basis where referral is not feasible. We implemented the guideline in program setting to increase access to treatment with high treatment success and low resultant mortality.

### Methods

Implementation research was conducted in four rural blocks of Lucknow district in Uttar Pradesh, India. It included policy dialogues with the central and state government and district level officials. A Technical Support Unit was established. Thereafter, capacity building across all cadres of health workers in the implementation area was done for strengthening of home based newborn care (HBNC) program, skills enhancement for identification and management of PSBI, logistics management to ensure availability of necessary supplies, monitoring and evaluation as well as providing feedback. Data was collected by the research team.

### Results

From June 2017 to February 2019 there were 24,448 live births in a population of 856106. We identified 1302 infants, aged 0–59 days, with any sign of PSBI leading to a coverage of

**Data Availability Statement:** All relevant data are within the manuscript and its Supporting Information files.

**Funding:** The study was supported by World Health Organization Geneva, Switzerland (World Health Organization: WHO www.who.int) via Project ID: MCA00415. The funders had role in study conceptualization and review and editing of the manuscript.

**Competing interests:** The authors have declared that no competing interests exist.

53% (1302/2445), assuming an incidence of 10%. However, in the establishment phase the coverage was 33%, while it was 85% in the implementation phase. Accredited social health activists (ASHAs) identified 81.2% (1058/1302) cases while rest were identified by families. ASHAs increased home visits within first 7 days of life in home based newborn care program from 74.3% (2781/3738) to 89.0% (3128/3513) and detection of cases of PSBI from 1.6% (45/2781) to 8.7% (275/3128) in the first and last quarter of the project, respectively. Of these 18.7% (244/1302) refused referral to government health system and 6.7% (88/1302) were treated in a hospital. Among cases of PSBI, there were 13.3% (173/1302) cases of fast breathing in young infant aged 7–59 days in whom referral was not needed. Of these 147 were treated by oral amoxicillin and 95.2% (140/147) were cured. Among those who needed referral, simplified treatment was given when referral was refused. There were 2.9% (37/1302) cases of fast breathing at ages of 0–6 days of which 34 were treated by simplified treatment with100% (34/34) cured;66.5% (866/1302) were cases of clinical severe infection of which 685 treated by simplified treatment with94.2% (645/685)cured and 09 died;17.3% (226/1302) cases of critical illness of which 93 were treated by simplified treatment, as a last resort, 72% (67/93) cured and 16 died. Among 255 cases who either did not seek formal treatment or sought it at private facilities, 96 died.

## Conclusion

Simplified treatment for PSBI is feasible in public program settings in northern India with good cure rates. It required system strengthening and supportive supervision.

## Introduction

India has been struggling tenaciously to tackle morbidity and mortality in children. In India, the Infant mortality rate per 1000 live births is 33 as compared to 41 in Uttar Pradesh, the most populous state. Neonatal mortality rate per 1000 live birth in India is 23 as compared to 30 in the state of Uttar Pradesh [1]. Leading infectious causes of mortality among young infants are pneumonia, sepsis and meningitis [2, 3]. With an estimated 9.8% percent case fatality risk of possible serious bacterial infection (PSBI) [4] in low resource setting, survivors are also at risk of long term disability [5]. Therefore, early diagnosis and treatment is crucial for saving lives of young infants.

Early identification of illness by caregivers in sick young infants is difficult due to non-specific signs. World Health organization (WHO) recommends postnatal home visits by healthcare providers to facilitate early identification of danger signs and promote care seeking [6]. Government of India launched the Home Based Newborn Care (HBNC) program in 2011 for accelerated reduction of neonatal mortality, especially in rural and remote areas where access to health care is challenging [7]. In the HBNC program, an Accredited Social Health Activist (ASHA) visits mothers and neonates at home at least six times within the first 42 days of birth and refer sick infants to the public health facility. WHO recommends sick young infants with PSBI be referred to a hospital for inpatient treatment with a seven to ten-day course of injection ampicillin or benzyl penicillin plus gentamicin [8, 9]. But hospitalization and life-saving treatment may not be accessible, acceptable or affordable to families in settings with high newborn mortality [10, 11, 12, 13].

In 2014, Ministry of Health and Family Welfare, Government of India issued guideline for Auxiliary Nurse and Midwife (ANMs)for management of sepsis in young infants where referral is not feasible i.e., use of injection gentamicin plus oral amoxicillin as a pre-referral dose or completion of treatment for 7 days [14].WHO, in 2015, published the guideline for management of PSBI in young infants when referral is not feasible [15] based on evidence from several countries [13, 16, 17, 18, 19], which recommended a simplified treatment regimen with oral amoxicillin plus injection gentamicin for treatment of clinical severe infection and severe pneumonia. In 2017, government of India issued an addendum with a revised list of signs for classifying young infants as PSBI, consistent with the WHO guideline, removing nasal flaring, grunting, 10 or more skin pustules and blood in stool and recommended treatment of fast breathing young infants 7–59 day of age with oral amoxicillin only [20].

In each village, HBNC program is being implemented with ASHA as its grass root functionary. In this program, an ASHA does home visits on specific days (Days 1, 3, 7, 14, 21, 28 and 42) with day 1 visit only for mothers who delivered at home. During each visit, in addition to post-natal care of mothers, they perform various activities to ensure well-being of young infants such as physical examination, temperature and weight recording, respiratory rate counting and counselling mothers for exclusive breast feeding etc. ASHA counsels mothers and looks for signs of illness in young infants including diarrhea and dehydration, jaundice, skin infections and PSBI [7]. All ASHA workers of a block are grouped in clusters of 20, each headed by an ASHA-*Sangini* who is responsible for supportive supervision and hand-holding in the field. Cluster meetings are organized monthly at the CHC for reporting routine activities to the manager and for ongoing skills enhancement.

ANMs are responsible for routine immunization at SC/PHC/CHC and administration of certain drugs/ treatment approved to community at SC [21]. They maintain records of pregnant women for ante-natal, deliveries, post-natal care and immunization of children in their area. All ANM in a block meet weekly and report their activities to the CHC medical officers and collect their supplies.

Despite the HBNC program going on from 2011 there is a high neonatal mortality rate in India, and 41% of deaths are due to pneumonia and sepsis [7]. Therefore, the rationale and primary objective of this implementation research was to evaluate acceptability, effectiveness and scalability of simplified treatment regimen to cases of PSBI when referral is not accepted in a HBNC program setting in Uttar Pradesh. This implementation research was part of a multisite study conducted at four sites, one each in four states in India. The current manuscript focuses on the findings from Lucknow site in the state of Uttar Pradesh. The other three other states where this study was conducted were- Haryana, Himachal Pradesh and Maharashtra.

## Study methodology

### Demographic details

Lucknow district in the state of Uttar Pradesh, India is located at an elevation of 123 meters above sea level. Uttar Pradesh has a crude birth rate of 25.9 per 1000 population [1]. Female literacy rate is 79.4% in women in 15–49 years' group. Lucknow district has an estimated population of 4,589,838 (Census 2011) of which 1,550,942(33.8%) live in the eight rural blocks [22]. Implementation research activities were conducted in four of the eight rural blocks, namely Mall, Gosainganj, Sarojini Nagar and Kakori, whose population is 856106 [23].

### Health infrastructure

Each village has an ASHA, a voluntary community-based female worker with at least eight years of formal education [24]. She works as a link between the community and public health

facilities, covers a population of about approximately 1000, an average village size. Each ASHA is provided with a HBNC kit which has bag, baby blanket, soap and soap case, functional weighing scales, digital thermometer and digital watch/timer to count respiratory rate. A sub-center (SC) has one ANM for a population of approximately 5000 to 8000 or 5–7 villages [21]. For every 6 SCs there is one Primary Health Center (PHC) covering a population of 30,000 to 50,000 with outpatient and in-patient facility of 6 beds and one delivery room. Each PHC has 1–2 medical doctor (either trained in modern medicine with MBBS degree and/or in alternative systems of medicine such as ayurveda, yoga therapy, unani and siddh medicine and homeopathy together called as AYUSH, three nurses, one ANM and a pharmacist [25].For 4 PHCs there is a Community Health Center (CHC) covering a population of 80,000 to 1,20,000 with outpatient and in-patient facility with 30 beds, a delivery room, operation theater, one sick newborn stabilizing unit with phototherapy and warmer units and a kangaroo mother care unit with beds for mothers with babies born with birth weight of less than 1800 g [26].CHC has one or two general doctors, few specialists one each being general physician, pediatrician, obstetrician, surgeon and anesthesiologist, seven nurses, one ANM, one public health midwife and one pharmacist. There is also facility for essential bed side laboratory investigations. All CHCs have government ambulance services for transport of patients, including pregnant women and young infants and are designated as the first referral units. Available public health infrastructure in Lucknow district and four implementation blocks is shown in **S1 Table**.

**Identification of cases of PSBI.**   During routine HBNC visits or additional visits, ASHA identifies sick young infants including those with PSBI signs. ASHAs advise parents of all such infants to visit the CHC/PHC/tertiary care hospitals for medical care. ASHA can consult the ANM, if she is available in the village on that particular day. When ANM confirms a case of PSBI, they are referred to a public hospital/ tertiary care facility after administration of pre-referral dose of injection gentamicin and oral amoxicillin (see **S2 Table**). The exception is for only fast breathing ($> 60$ breaths per minute) in 7–59 days old infants who are given 7 days of oral amoxicillin without referral. Similar process is carried out by attending physician for young infants with PSBI signs who present at a PHC/CHC. If referral is not accepted, then the sick infant is categorized into critical illness (convulsions, no movement on stimulation, not feeding at all) or clinical severe infection (less than normal movements or movement only on stimulation, not feeding well, severe chest indrawing, temperature of $> = 38°C$ or $<35.5°C$) or severe pneumonia (fast breathing 0–6 days). WHO recommended simplified treatment is offered for clinical severe infection and severe pneumonia and referral is reinforced in those with critical illness as they are unable to take oral antibiotics [15].

**Simplified treatment.**   Oral amoxicillin twice a day (40–50 mg/dose) plus intramuscular injection gentamicin once a day (7 mg/Kg) for 7 days was administered to those with clinical severe infection or severe pneumonia [14]. Parents either brought the case daily to the health facility to receive injectable gentamicin or ANM visited home of the case daily for administering it. Parents gave the first dose of oral amoxicillin under medical supervision and then subsequent doses at home. ASHA visited such cases almost daily to ensure adherence to treatment. Young infants aged 7–59 days with only fast breathing were given only oral amoxicillin twice a day for 7 days [14]. Cases were categorized as "treatment success" when the care providers perceived that the infant was cured or better on day 8 of follow up and as "treatment failure" when condition worsened or there were persistent signs on day 8 or the patient died.

**Policy dialogue.**   For informing policy makers and for their approval for implementation research work, national as well as state level meetings were held prior to initiation of the project as the implementation research was done by the state government and health care functionaries. There was a continuous communication of the investigators with the implementers and the policy makers.

**National level meeting.** There were two meetings held at national level. First meeting was held in October 2016. It was attended by National advisors to government, representatives from Central Ministry of Health and family welfare, investigators of the four participating states, the World Health Organization representatives and expert advisors from Save the Children. Outcome of the meeting was adaptation of guidelines for PSBI management in young infants where hospitalization is not feasible [14]. The second meeting was held with all the stakeholders who were present in the first meeting in New Delhi in July 2017. During this meeting, plans of all the four research sites were reviewed and discussed. Suggestions and recommendations to and from the government were incorporated customized to every research site.

**State level meeting.** A meeting of investigators with the Director, managing director and general manager Child health of Uttar Pradesh-National Health Mission was held in January 2017. They were briefed about the implementation research. All the officials appreciated and supported it and assured the availability of Sick newborn care unit and Kangaroo Mother Care units in four implementation blocks for management of the referred sick young infants. The director proposed incentives of INR 100 to the ASHA workers on successful referral of cases of PSBI to public health facility. He also recommended a brief re-training, especially for ASHA workers on management of PSBI. A technical advisory group (TAG) was formed to support and guide the research. Its members were representatives of state government and multilateral agencies working in child health such as UNICEF and Save the Children. Functions of the TAG was to review program implementation once in 6 months and provide inputs. At the end of the implementation phase, a state level dissemination meeting was also held where members of TAG were invited.

**Role of Technical Support Unit (TSU).** A TSU was established at KGMU which had the investigators and research staff trained by the investigators for supportive supervision of ASHAs and ANMs; effective implementation of HBNC program and simplified treatment for PSBI when referral was not feasible; necessary data collection from health care providers' records for continuous evaluation; and provide feedback to the program for quality improvement and for assessing effectiveness of the implementation research.

**Role of district health staff.** Government doctors conducted trainings for strengthening home based newborn care program and giving simplified treatment, logistic support and ensured supplies as well as continued their routine monitoring and supervisions. ASHA and ANMs conducted their routine activities which now included identification and management of cases of PSBI by simplified treatment where referral was refused.

## Ethical approval

The study was approved by the Institutional Ethics committee of King George Medical University, Lucknow, Uttar Pradesh and Ethics Review Committee of World Health Organization, Geneva.

## Phases of the project

The project had two distinct phases: Establishment phase and Implementation phase.

**Establishment phase (June 2017 to Feb 2018).** The activities during this phase were:

a. **Training:** Initial training of master trainers for PSBI was conducted centrally in New Delhi (November 2016) by the WHO facilitators using pre-validated materials [15, 27]. The master trainers trained pediatricians and some medical officers in the state public health system for them to become the trainers for medical officers, nurses, ANMs, ASHAs and ASHA

"*Sangini*". The training emphasized on the acquisition of necessary skills on identification and management of PSBI or neonatal sepsis in young infants (**S3 Table**). The research staff hired were trained by project investigators on HBNC and PSBI over three days, using the modules provided by the WHO which were used for the training for trainers. In addition, they were taken to the neonatal intensive care unit and sick newborn care unit of King George's Medical University and shown all signs of PSBI at least twice over a period of one week by the project investigator. There were multiple transfers of doctors and ANMs during the project. One to one training of new doctors and ANMs with refresher training of those trained earlier was conducted by TSU staff at their place of posting, which mandated by the government that they must remain in their places of duty all day. Newly inducted ASHA were also trained at CHCs (**S3 Table**).

b. **Baseline survey:** This survey was conducted in 120 villages of four implementation rural blocks of Lucknow using the thirty cluster sampling technique. In each block, 30 villages were included based on the population enumerated in census 2011 [1]. The villages were selected on the basis of size as small villages (<200 households), medium sized villages (200–400 households) and large sized villages (>400 households) in the ratio of 4:3:1 respectively. Mothers of infants 0–59 days was the unit if this survey. Mothers of all the young infants (0–59 days' age) registered with the ASHA in selected villages were visited and interviewed after taking informed written consent. Using a predesigned, structured questionnaire they were asked about the number of HBNC visits done for their young infant and what the ASHA did on each visit. ASHAs of selected villages were interviewed using a predesigned, structured questionnaire and in addition one ASHA who was doing HBNC on the day of survey was interviewed and observed as she conducted the procedures laid down in HBNC. All the ASHA were interviewed to assess training given to them and their knowledge about HBNC and the functional status of HBNC kit given to them was evaluated. A predesigned, structured interview questionnaire was used.

During this survey, to assess the knowledge of HBNC, recognition and management of PSBI, in-depth qualitative interview of a few mothers, ANMs and doctors was also done by purposive sampling. Baseline audit of health facilities was done to check for availability of oral amoxicillin, injectable gentamicin, 1 ml syringes and gentian violet lotion.

c. **Data Collection for process indicators:** From the records maintained by the district health staff, data was abstracted by TSU on numbers of pregnant women and deliveries per village, number of times HBNC visit was done for each infant with findings of each visit like weight, respiratory rate, temperature and presence of any danger signs or signs of PSBI. In about 15% of such visits the TSU staff observed the practice of ASHA workers and identified areas which needed strengthening either with support of the ASHA-"*sangini*" or through re-training. The district health continued usual independent monitoring and evaluation of the HBNC program.

The TSU staff also collected information about cases of PSBI with, their presenting symptoms from the records maintained by the district health staff. Each PSBI case was visited by the research staff to validate findings abstracted as well as collect information on place of treatment, type of treatment specifically simplified treatment when referral was not accepted, response to administered treatment on days 4 and 8. Data on cases of PSBI on simplified treatment was collected by the research staff but in cases where parents took their child to a private provider or accepted referral to a district hospital, follow-up at home could not be done.

d. **Refresher training and support:** Members of TSU attended routine monthly cluster meetings of ASHAs and ANMs in their respective blocks. During this time, they scheduled refresher training on HBNC and PSBI program for ASHA with medical officers or ASHA "*sangini*" and observed them. During such trainings, skills which were found to be deficient like respiratory rate counting, physical examination to identify signs of infection and counselling for referral were re-enforced. ANMs were motivated by medical officers to administer pre-referral dose and in providing treatment at home/SC. Experiences on successful case identification and/or referral were also shared to encourage others. Medical officers appreciated ASHA who identified infants with danger signs and referred them for management. These meetings were also utilized for dialogue and fostering linkages between ASHA and their respective ANMs (See **S4 Table**).

e. **Management Information System (MIS):** Common case report forms were developed and customized in India for all the sites for research data collection. It included five forms: 1) Pregnancy identification and listing form, 2) Postnatal Home visit record, 3) Sick Young infant assessment form, 4) Facility Follow-up form (for record of Simplified Treatment given), 5) Home follow up and outcome. Information collected was entered in the secure web based application, Research Electronic Data Capture "RED Cap" [28]. Information collected from the field was checked and verified before entry into the database. Data analysis was done using MS excel and SPSS version 18.0 [29]. Univariate distribution of variables was done and presented as proportion with 95% confidence interval where required. Since it was an implementation research no inferential statistical analysis was done.

The establishment phase took around 9 months to stabilize and deliver optimal performance with reference to number of cases of sick infants identified and administering of pre-referral dose and/or simplified treatment.

**Implementation phase (Mar 2018 to Feb 2019).** After the establishment phase, the intervention activities continued in the implementation mode, performing same activities as done in the establishment phase. The only new activity done during this phase was midline survey.

a. **Midline Survey:** One year after baseline survey, another survey was done using the same methodology as baseline survey. Same villages were visited as done in baseline survey however, mothers having infants 0–59 days were new. If there was no mother in the village selected earlier, adjoining village of the same size was visited. During this survey, interview of mothers was done to evaluate the change in knowledge level. ASHA interview and observation was also done to evaluate the change in her skills. Interview of medical officers and ANMs was done to evaluate the change in their knowledge for treatment of young infants and also to have a feedback for the simplified treatment of PSBI.

## Results

### Study characteristics

This implementation research was conducted in four rural blocks (Gosainganj, Kakori, Mall and Sarojini Nagar) of Lucknow district. The population of these four blocks is 856,106. During the study period from 01 June 2017 to 28 Feb 2019, 25,028 pregnant women were registered with ANMs in 376 villages of the four participating blocks. Among these there were 24,448 registered live births.

**Baseline survey.** Baseline survey had both qualitative as well as a quantitative component and was conducted in 120 selected villages of four implementation blocks from February 2017 to May 2017. As a part of qualitative data collection, in-depth interviews (IDIs) were

conducted with mothers of infants aged 0–59 days, ANMs and doctors. Twenty eligible mothers (five per block) were purposively selected for IDIs and interviewed for knowledge of HBNC program and danger signs of illness in young infants. Mean age of mothers who participated was 25.17±3.52 years.

During IDIs, it was found that most of the mothers did not know the detailed schedule of visit of ASHAs during HBNC program. However, they informed that the ASHA visited the home 3–4 times within 42 days of the birth of a neonate. Almost all mothers reported that ASHAs did not tell them about signs of sickness in young infants. She usually gave instructions for breast-feeding, vaccinations and maintaining temperature of the child. Most of the mothers were not aware of all the danger signs of sickness. Some recalled fever, loose motions and fast breathing as signs of sickness. One of the respondent said *"Whenever ASHA visits, she asks about the health of the infant, washes and air-dries her hands, weighs and measures the temperature of the infant"* [Block–Kakori]

IDIs were also conducted with 24 ANMs (6 per block and one per SC) at their respective SC. Purpose of conducting IDI was to obtain information on their training on HBNC, knowledge of danger signs of illness in young infants and its management. During IDIs, the ANMs informed that all of them were trained in HBNC program at the time of appointment by the government. HBNC training had a larger knowledge component and relatively small skills development component. There was no in-service reinforcement of training. None of them recalled having seen any sick young infant with "danger signs" in the last one month. Most of them said that fast breathing, loose motions, fever, skin infections and jaundice were danger signs. They referred such cases to a doctor to a CHC, which often had a pediatrician. They were not aware about giving pre-referral anti-biotics or any other advice at the time of transportation. When they were asked that whether they were comfortable in giving injectable antibiotic treatment, most of the ANMs were apprehensive that if an infant died or disease progressed the family members would blame them.

IDIs were conducted of four medical doctors (one per block), who were not pediatricians, at their respective CHC. Respondent was purposively selected to assess their knowledge about symptoms of PSBI and its management. During IDIs, medical doctors informed that at CHC/PHC that was without pediatricians, only some of the doctors gave treatment to sick young infants. Most enumerated fever, diarrhoea, skin infections, fast breathing, lethargy, inability to feed and jaundice as danger signs and referred sick neonates to higher centers without any pre-referral dose. They also informed that most of the families did not accept referral to higher government hospitals and went to private hospital for treatment.

Baseline health facility audit showed that there was shortage of oral amoxicillin suspension in most facilities and amoxicillin dispersible tablets were not available. Vials of gentamicin injections were available in the pharmacy of all health facilities but not issued to the ANMs for use in the field for management of sick young infants. One ml injections with 26 gauze syringes were not present in the government supply.

Quantitative structured interviews were conducted with mothers who did not participate in the qualitative interview. All 416 eligible mothers were interviewed from 120 villages. Sociodemographic characteristics of these mothers are given in **Table 1**.

In addition to interview of 416 mothers, 118 ASHAs were also interviewed using a structured questionnaire. Knowledge of the mothers and ASHAs elucidated about danger signs during baseline survey is also shown in **Table 2**.

**Midline survey.** Midline survey was conducted in 120 villages of four implementation block from June 2018 to July 2018. While 51.6% (62/120) villages were the same as in baseline survey.48.3% (58/120) were new since there was no young infant in the corresponding number of villages at the time of midline survey. Structured interviews of 266 eligible mothers and 120

**Table 1. Socio-demographic characteristics of the parents during baseline and midline surveys.**

|  | Baseline | Midline | P Value |
|---|---|---|---|
|  | N = 416 | N = 266 |  |
| Age in years of mothers (Mean age ± SD) | 25.4 ± 3.52 | 24.93 ± 3.7 | 0.10 |
|  | n, % | n, % |  |
| Uneducated Mothers | 116, 27.9 | 64, 24.1 | 0.31 |
| Occupational status of the mothers |  |  |  |
| Housewives | 399, 95.9 | 262, 98.5 | 0.007 |
| Daily wagers | 13, 3.1 | - | 0.004 |
| Salaried / small business | 04, 1 | 1,0.4 |  |
| Age in years of fathers (Mean age ± SD) | 29.1 ± 4.83 | 28.25 ± 4.526 | 0.020 |
| Uneducated Fathers | 81, 19.5 | 42, 15.8 | 0.27 |
| Occupational status of fathers |  |  |  |
| Daily Wager | 262, 63.0 | 165, 62.2 | 0.91 |
| Farmer/ Small Business | 116, 27.9 | 85, 32.1 | 0.28 |
| Salaried (Clerical/Professional) | 38, 9.1 | 15, 5.6 | 0.13 |
| Age of the Index Child |  |  |  |
| ≤7 days | 49, 11.8 | 22, 8.3 | 0.06 |
| 8-59days | 366, 88.0 | 242, 91.0 |  |
| Age in days of index child (Mean Age± SD) | 28.1 ± 15.96 | 32.53 ± 16.48 | 0.0006 |
| Pregnant women made antenatal visits |  |  |  |
| ≥ 4 | 267,64.2 | 198,74.4 | 0.008 |

**Table 2. Change of knowledge of mothers and ASHA about danger signs between baseline and midline's survey.**

|  | Baseline | Midline | P Value |
|---|---|---|---|
| **% of mothers who were visited by ASHA aware of danger signs in young infants** | N = 416 | N = 266 |  |
|  | n, % | n, % |  |
| Not feeding at all/Not feeding well | 137, 47.1 | 240, 94.5 | < 0.0001 |
| Convulsions | 47, 16.2 | 181, 73 | < 0.0001 |
| Severe chest in drawing | 97, 33.3 | 226, 89.0 | < 0.0001 |
| Hot to touch | 149, 51.2 | 242, 95.3 | < 0.0001 |
| Cold to touch | 56, 19.2 | 232, 91.3 | < 0.0001 |
| Fast Breathing | 86, 29.6 | 222, 87.4 | < 0.0001 |
| Less than normal movements or movement only on stimulation | 93, 21.6 | 194, 76.4 | < 0.0001 |
| Do not know/ No information | 66, 22.7 | 20, 7.9 | < 0.0001 |
| **% of ASHAs aware of danger signs in young infant** | N = 118 | N = 120 | P Value |
|  | n, % | n, % |  |
| Not feeding at all/Not feeding well | 71,65.1 | 118, 98.3 | <0.0001 |
| Convulsions | 41, 37.6 | 112, 93.3 | <0.0001 |
| Severe chest in drawing | 61, 56.0 | 118, 98.3 | <0.0001 |
| Hot to touch | 64,58.7 | 120, 100 | <0.0001 |
| Fever; >99˚F | 37,31.0 | 117, 97.5 | <0.0001 |
| Cold to touch | 36, 33.0 | 117, 97.5 | <0.0001 |
| Hypothermia;; <97.5˚F | 88, 74.6 | 114, 95.0 | <0.0001 |
| Less than normal movements or movement only on stimulation | 39, 35.8 | 104, 86.7 | <0.0001 |
| Fast breathing (breaths 60/min or more) | 75, 68.8 | 117, 97.5 | <0.0001 |

ASHAs were done to assess the change in their knowledge about danger signs. The sociodemographic characteristics of these mothers are shown in **Table 1.** Change in knowledge of mother and ASHAs about individual danger signs is given in **Table 2**. There was statistically significant improvement in knowledge as compared to baseline interview.

IDIs were conducted with 16 ANMs (04 ANMs per CHC) to understand the experiences in the implementation research, also to receive their suggestions for strengthening management of PSBI. Most of the ANMs had identified a sick young infant and more than half of them had administered either pre-referral dose or simplified treatment to cases of PSBI. Most of the ANMs appreciated the availability of treatment at SC/PHC/CHC. They said that it was easy to deliver simplified treatment and most of the ANMs had developed confidence in giving the gentamycin injection in the field setting. Villagers now held them in esteem as they perceived that their treatment had saved lives of sick young infants. Most of the ANMs requested for retraining sessions of ASHA and ANMs. One of the ANM, sharing her experience of implementation research that "*The revision training on PSBI identification and its management should continue and the availability of medicines should be ensured*" [Block-Mall]

In-depth interviews of 24 non-paediatreic doctors at CHC/PHC was taken to assess feasibility of continuing simplified treatment at SC/PHC/CHC to cases of PSBI after implementation program was over. Most of the non-paediatric doctors acknowledged the improvement in their knowledge and skills to effectively treat sick young infants with danger signs. They also said that they were able to give treatment of PSBI case at CHC/PHC and referred only those with critical illness to higher centres.

The medical doctors acknowledged that there was improvement in skills of ASHAs and ANMs in identifying and referring sick young infants with PSBI. Most of them said that simplified treatment to cases of PSBI would be integrated in routine practice and continue beyond the implementation research period. However, periodic refresher training would be beneficial.

One of the non-pediatric doctor shared his experience of implementation research as follows "*We would continue the simplified treatment as this is very simple, safe and easily administrable at CHC/PHC/SC level*" [Block- Sarojini Nagar]

## Results of the implementation

In a population of 856106, 24,448 live births were recorded from June 2017 to February 2019. Based on 10% incidence rate of PSBI, 2445 infants of age 0–59 days were expected to have any sign of PSBI. We identified 1302 infants of age 0–59 days with any sign of PSBI leading to coverage of 53% (1302/2445). However, in the establishment phase the coverage was 33%, while it was 85% in the implementation phase.

As a result of supportive supervision of the ASHAs in the HBNC program, identification of PSBI and fast breathing cases improved in each successive quarter (**Table 3**)**.**

Of the 1302 cases of PSBI, 1058 (81.3%) were identified by ASHA and 244 (18.7%) by families (**Fig 1**)**.** Of 1003 brought to a PHC/CHC, 812 refused referral to a hospital but accepted simplified antibiotic treatment. Eighty-eight young infants (33 accepting referral to a hospital from PHC/CHC and 55 brought directly by families) were treated at a district hospital. Parents of 58 (4.5%)young infants took either used home remedies, sought care from faith healers or didn't receive any formal treatment, whereas parents of 197 (15.2%) young infants sought treatment from private practitioners (11 initially went to government facility but refused treatment there).

Details of identification and classification of illness are given in **Table 4**. Day 4 and 8 follow-up was done in 93.6% (898/959) cases of PSBI on simplified treatment. Those lost to follow up were largely due to migration of families.

**Table 3. Home based newborn care visits, identification and management of possible serious bacterial infection cases by establishment and implementation phases.**

| | Phases I: Establishment | | | Phase II: Implementation | | | |
| | (Jun17- Feb18) | | | (Mar18-Feb19) | | | |
| | Quarter 1 (Jun17-Aug17) | Quarter 2 (Sep17-Nov17) | Quarter 3 (Dec17-Feb18) | Quarter 1 (Mar18-May18) | Quarter 2 (Jun18-Aug18) | Quarter 3 (Sep18-Nov18) | Quarter 4 (Dec18-Feb19) |
|---|---|---|---|---|---|---|---|
| Live births | 3738 | 3691 | 3124 | 3140 | 3409 | 3833 | 3513 |
| Home Deliveries n, (%) | 277, (7.4) | 324, (8.8) | 256, (8.2) | 247, (7.9) | 244, (7.1) | 314, (8.2) | 276, (7.8) |
| Day 1 for home deliveries n, (%) | 163, (58.8) | 217, (67.0) | 166, (64.8) | 168, (68.0) | 187, (76.6) | 221, (70.4) | 214, (77.5) |
| PNV day 3/ Visit 2 n, (%) | 2255, (60.3) | 2614, (70.8) | 2057, (65.8) | 2209, (70.3) | 2391, (70.1) | 2859, (74.6) | 2637, (75.1) |
| PNV day 7/ Visit 3 n, (%) | 2416, (64.6) | 3058, (82.8) | 2398, (76.8) | 2603, (82.9) | 2733, (80.2) | 3445, (89.9) | 3320, (94.5) |
| PNV day 14/ Visit 4 n, (%) | 2157, (57.7) | 3095, (83.8) | 2466, (78.9) | 2626, (83.6) | 2718, (79.7) | 3597, (93.8) | 3159, (89.9) |
| PNV day 21/ Visit 5 n, (%) | 1836, (49.1) | 3098, (83.9) | 2510, (80.3) | 2638, (84.0) | 2606, (76.4) | 3624, (94.5) | 3150, (89.7) |
| PNV day 28/ Visit 6 n, (%) | 1556, (41.6) | 3072, (83.2) | 2524, (80.8) | 2568, (81.8) | 2536, (74.4) | 3649, (95.2) | 3151, (89.7) |
| PNV day 42/ Visit 7 n, (%) | 1132, (30.3) | 2991, (81.0) | 2541, (81.3) | 2533, (80.7) | 2364, (69.3) | 3559, (92.8) | 3180, (90.5) |
| Live births visited by ASHA in 1st week of life n, (%) | 2781, (74.3) | 2950, (79.9) | 2545, (81.4) | 2618, (83.3) | 2918, (85.5) | 3376, (87.8) | 3128, (89.0) |
| PSBI Identified n, (%) | 45/ 2781, (1.6) | 127/2950, (4.4) | 106/2545, (4.1) | 261/2618, (9.9) | 239/2918, (8.1) | 249/3376, (7.3) | 275/3128, (8.7) |
| Fast Breathing on at 7–59 days of age n, (%) | - | 17/127, (13.3) | 30/106, (28.3) | 39/261, (14.9) | 22/239, (9.2) | 46/249, (18.4) | 56/275, (20.3) |
| Simplified treatment n, (%) | 30/45, (66.7) | 100/127, (78.7) | 81/106, (76.5) | 195/261, (74.7) | 170/239, (71.2) | 188/249, (75.5) | 195/275, (70.9) |
| Pre-referral Dose of injection gentamicin | 0/2 | 1/5 | 1/1 | 0/3 | 4/7 | 4/5 | 6/10 |
| Refused referral and accepted simplified treatment n, (%) | 30/45, (66.7) | 100/127, (78.7) | 81/106, (76.5) | 195/261, (74.7) | 170/239, (71.2) | 188/249, (75.5) | 195/275, (70.9) |
| Accepted referral and went to tertiary care hospital n, (%) | 05/45 (11.1) | 09/127 (7.0) | 01/106 (0.9) | 09/261 (3.4) | 24/239 (10.0) | 16/249 (6.4%) | 24/275 (8.7) |
| Refused referral and treatment at public health facility n, (%) | 10/45 (22.2) | 18/127, (14.2) | 24/106, (22.7) | 57/261, (21.9) | 45/239, (18.9) | 45/249, (18.1) | 56/275, (20.3) |

Abbreviation: PSBI–Possible serious bacterial infection

PNV–Post natal Visit

The proportion of identified PSBI increased from 1.6% in first quarter to 8.7% in the last quarter.

Details of compliance and outcome of cases of PSBI who treated with Simplified treatment and who accepted referral and were treated at district hospital are given in **Table 5**. Reasons for low compliance to simplified treatment could be either family stopped treatment due to improvement in sickness or health centre being far away or family opting for treatment elsewhere. Since injectable gentamicin was not available in the government hospitals all the time, it was replaced by injection amikacin. Hence compliance to all seven injections of gentamicin was low (132/812, 16.3%).

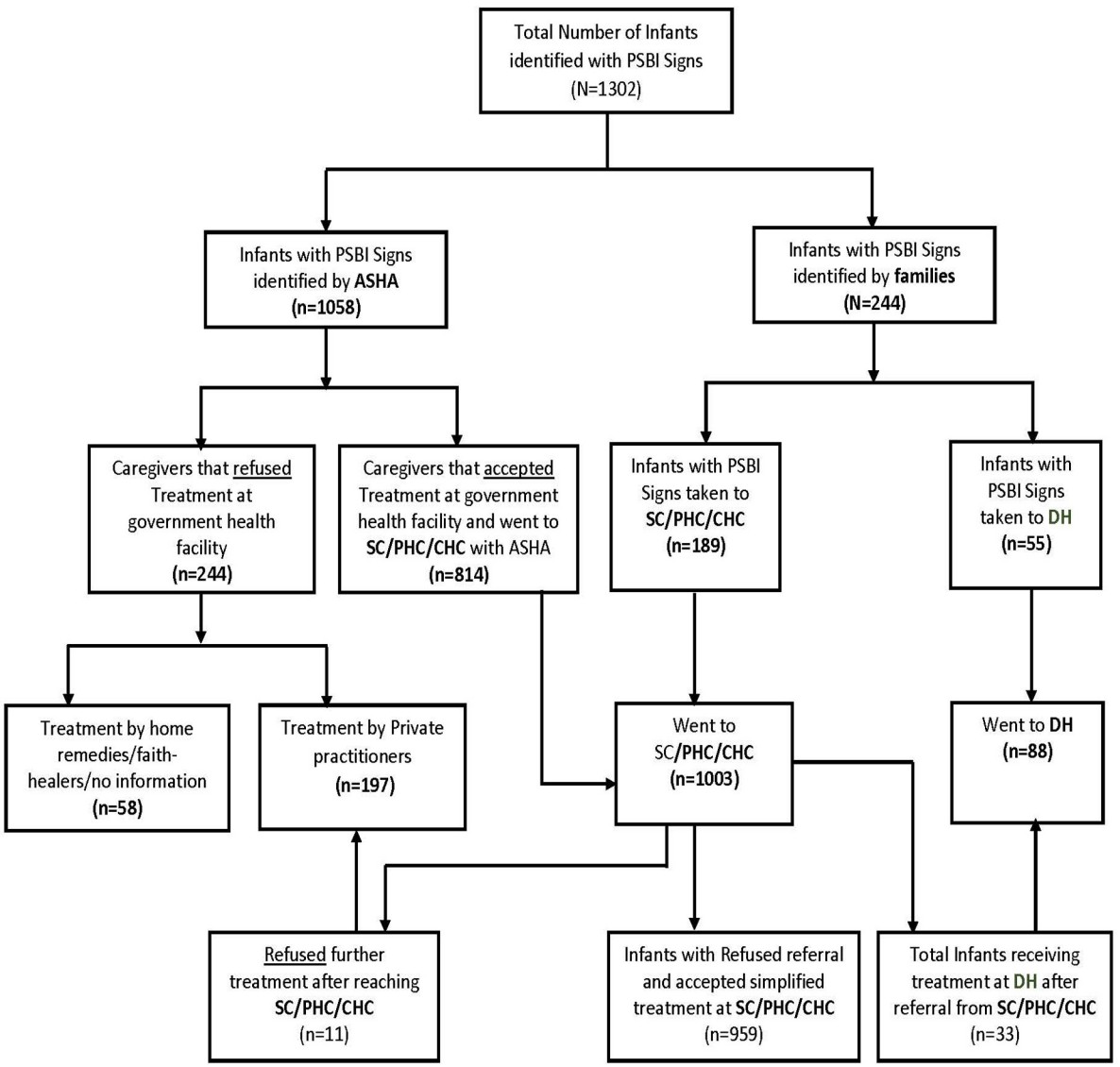

SC = Sub Centre, PHC = Primary Health Centre, CHC = Community Health Centre, DH = District Hospital; FB=Fast Breathing

**Fig 1. Places of identification and management of cases of possible serious bacterial infections.**

Of 1302 cases of PSBI, 37 (2.8%) had fast breathing in0-6 days of age young infants, 173 (13.3%) had fast breathing in7-59days of age young infants, 866 (66.5%) had clinical severe infection signs and 226 (20.4%) had critical illness.

Mandatory day 4 follow up was carried in 95.8% (919/959) cases who accepted simplified treatment. Only 13.7% (131/959) received full7 days of WHO recommended simplified treatment course. Of those who accepted simplified treatment at PHC/CHC, 12.7% (122/959) received other oral antibiotics primarily due to non-availability of oral amoxicillin in health facilities; 512/812 (63.1%) who needed injection gentamicin received it, where the rest were treated by injection amikacin, which was available at health facilities. Of the 812 cases of PSBI, who received injectable treatment, injection was given by medial officer in 87.8% (713/812), by ANM in 10.5% (86/812) and by Nurse in 1.6% (13/812) cases. ANMs administered injectable

**Table 4. Identification and classification in young infants (0–59 days) with possible serious bacterial infection.**

| Parameters | All PSBI cases | Fast Breathing 7–59 days | Fast Breathing 0–6 days | Clinical Severe Infection 0–59 days | Critical Illness 0–59 days |
|---|---|---|---|---|---|
| A. **Identification and Referral History** | N = 1302 n/m (%) | N = 173 n/m (%) | N = 37 n/m (%) | N = 866 n/m (%) | N = 226 n/m (%) |
| **Identified by Families and brought to Sub-centre/ Primary Health Centre/ Community Health Centre** | 189/1302 (14.5) | 21/173 (12.1) | 18/37 (48.6) | 103/866 (11.9) | 47/226 (20.8) |
| **Identified by families and brought to District Hospital / Teaching Hospital** | 55/1302 (4.2) | 1/173 (0.6) | 0/37 (0) | 38/866 (4.4) | 16/226 (7.1) |
| **Identified by ASHA** | 1058/1302 (81.3) | 151/173 (87.3) | 19/37 (51.3) | 725/866 (83.7) | 163/226 (72.1) |
| a) Went to Subcentre/ Primary Health Centre/ Community Health Centre | 814/1058 (76.9) | 127/151 (84.1) | 17/19 (89.5) | 599/725 (82.6) | 71/163 (43.6) |
| b) Refused referral to government health system | 244/1058 (23.1) | 24/151 (15.9) | 2/19 (10.5) | 126/725 (17.4) | 92/163 (56.4) |
| i)Went to Private for treatment | 186/244 (76.2) | 24/24 (100) | 2/2 (100) | 110/126 (87.3) | 50/92 (54.3) |
| ii)No formal treatment received* | 58/244 (23.8) | 0/24 (0) | 0/0 (0) | 16/126 (12.7) | 42/92 (45.7) |
| **Total infants assessed at Subcentre/ Primary Health Centre/ Community Health Centre** | 1003/1302 (77.0) | 148/173 (85.5) | 35/37 (94.6) | 702/866 (81.1) | 118/226 (52.2) |
| Referred to District Hospital | 33/1003 (3.2) | 0/148 (0) | 1/35 (2.9) | 12/702 (1.7) | 20/118 (17) |
| Treated as outpatient on simplified treatment that does not need referral | 147/1003 (14.7) | 147/148 (99.3) | - | - | - |
| Refused referral to a hospital and accepted simplified treatment | 812/1003 (81) | | 34/35 (97.1) | 685/702 (97.6) | 93/118 (78.8) |
| Refused treatment at Primary Health Centre/ Community Health Centre | 11/1003 (1.1) | 1/148 (0.7) | 0/0 (0) | 5/702 (0.7) | 5/118 (4.2) |
| B. **Follow up** | N = 959 n/m (%) | N = 147 n/m (%) | N = 34 n/m (%) | N = 685 n/m (%) | N = 93 n/m (%) |
| **Infants followed up on day 4 (*mandatory day as to WHO guideline*)** | 919/959 (95.8) | 146/147 (99.3) | 34/34 (100) | 663/685 (96.8) | 76/93 (81.7) |
| Infants completed all follow-up visits on Day 4 and Day 8 | 898/959 (93.6) | 142/147 (96.6) | 34/34 (100) | 651/685 (95.0) | 71/93 (76.3) |
| Infants partially followed-up (*all follow-up visits not completed*) | 59/959 (6.2) | 5/147 (3.4) | 0/34 (0.0) | 33/685 (4.8) | 21/93 (22.6) |
| No of infants lost to follow-up (final outcome unknown) | 2/959 (0.2) | 0/147 (0.0) | 0/34 (0.0) | 1/685 (0.1) | 1/93 (1.1) |
| C. **Simplified Treatment given at first level health facility (Subcentre/ Primary Health Centre/ Community Health Centre)** | N = 959 n/m (%) | N = 147 n/m (%) | N = 34 n/m (%) | N = 685 n/m (%) | N = 93 n/m (%) |
| No of infants completed treatment for 7 days | 131/959 (13.7) | 105/147 (71.4) | 16/34 (47.1) | 81/685 (11.8) | 34/93 (36.6) |

* Home remedies, faith healers or no formal treatment

treatment to 44 cases in establishment and 42 in implementation phase. This treatment was administered by 34.7% (41/118) ANMs. Overall, 92.4% (885/959) PSBI cases who accepted simplified treatment were successfully treated, whereas 2.6% (n = 25) died, 9 with signs of clinical severe infection and 16 with critical illness. Of the 88 cases treated at district hospital, 67 (76.1%) were cured, 11 (12.5%) died, 8 with signs of critical illness and 3 with clinical severe infection.

Of the 255 PSBI cases who refused treatment from government facilities, 25.4% (50/197) who sought care from private providers were reported dead on follow-up visits and 79.3% (46/58) who didn't receive any formal treatment died within 7 days of becoming sick. Death due to birth asphyxia or prematurity could not be ruled out in these cases.

During implementation research we faced several challenges and found solutions to many of them in collaboration with the public health authorities **Table 6.**

**Table 5. Compliance and outcome of cases of possible serious bacterial infection who treated with simplified treatment and who accepted referral and were treated at district hospital.**

| Parameters | All PSBI cases | Fast Breathing 7–59 days | Fast Breathing 0–6 days | Clinical Severe Infection 0–59 days | Critical Illness 0–59 days |
|---|---|---|---|---|---|
| A. **Compliance of treatment** | N = 959 n/m (%) | N = 147 n/m (%) | N = 34 n/m (%) | N = 685 n/m (%) | N = 93 n/m (%) |
| **Number of oral amoxicillin doses** 14 Doses | 517/959 (53.9) | 105/147 (71.4) | 31/34 (91.2) | 333/685 (48.6) | 48/93 (51.6) |
| 10–13 Doses | 293/959 (30.6) | 41/147 (27.9) | 3/34 (8.8) | 228/685 (33.3) | 21/93 (22.6) |
| 6–9 Doses | 10/959 (1.0) | 1/147 (0.7) | - | 9/685 (1.3) | - |
| 5 or less Doses | 17/959 (1.8) | - | - | 9/685 (1.3) | 8/93 (8.6) |
| Other oral antibiotics received§ | 122/959 (12.7) | - | - | 106/685 (15.5) | 16/93 (17.2) |
| **Injection gentamicin received¶** | N = 812 n/m (%) | NA | N = 34 n/m (%) | N = 685 n/m (%) | N = 93 n/m (%) |
| 7 injections | 132/812 (16.3) | NA | 16/34 (47.1) | 80/685 (11.7) | 36/93 (38.7) |
| 5–6 injections | 217/812 (26.7) | NA | 10/34 (29.4) | 192/685 (28.0) | 15/9 (16.1) |
| 3–4 injections | 90/812 (11.1) | NA | 5/34 (14.7) | 80/685 (11.7) | 5/93 (5.4) |
| 1–2 injections | 73/812 (9.0) | NA | - | 52/685 (7.6) | 21/93 (22.6) |
| Other injectable antibiotic (amikacin) received∞ | 300/812 (36.9) | NA | 3/34 (8.8) | 281/685 (41.0) | 16/93 (17.2) |
| B. **Outcome for those who accepted simplified treatment (based on respondent's perception)** | N = 959 n/m (%) | N = 147 n/m (%) | N = 34 n/m (%) | N = 685 n/m (%) | N = 93 n/m (%) |
| Infants declared as 'Treatment success' | 886/959 (92.4) | 140/147 (95.2) | 34/34 (100) | 645/685 (94.2) | 67/93 (72.0) |
| Infants declared as 'Treatment failure' | 73/959 (7.6) | 7/147 (4.8) | 0/34 (0) | 40/685 (5.8) | 26/93 (28.0) |
| **Reason for treatment failure** | N = 73 n/m (%) | N = 7 n/m (%) | N = 0 n/m (%) | N = 40 n/m (%) | N = 26 n/m (%) |
| Left treatment | 45/73 (62.5) | 7/7 (100) | 0/0 (0) | 30/40 (75) | 8/26 (30.8) |
| Persistence of presenting sign on day 8 | 1/73 (1.4) | 0/7 (0) | 0/0 (0) | 0/40 (0) | 1/26 (3.8) |
| Deaths | 25/73 (34.2) | 0/7 (0) | 0/0 (0) | 09/40 (22.5) | 16/26 (61.5) |
| Outcome unknown | 2/73 (2.8) | 0/7 (0) | 0/0 (0) | 01/40 (2.6) | 1/26 (3.8) |
| C. **Outcome of the illness on Day 7 of those who accepted referral and were treated at district hospital** | N = 88 n/m (%) | N = 1 n/m (%) | N = 1 n/m (%) | N = 50 n/m (%) | N = 36 n/m (%) |
| Cured | 67/88 (76.1) | 1/1 (100) | 1/1 (100) | 45/50 (90) | 20/36 (55.6) |
| Better | 7/88 (8.0) | 0/1 (0) | 0/1 (0) | 02/50 (4) | 05/36 (13.9) |
| Still sick | 2/88 (2.3) | 0/1 (0) | 0/1 (0) | 0/50 (0) | 02/36 (5.6) |
| Same | 1/8 (1.1) | 0/1 (0) | 0/1 (0) | 0/50 (0) | 01/36 (2.8) |
| Outcome unknown | 0/88 (0) | 0/1 (0) | 0/1 (0) | 03/50 (6) | 0/36 (0) |
| Deaths | 11/88 (12.5) | 0/1 (0) | 0/1 (0) | 03/50 (6) | 08/36 (22) |

¶ Excluding cases of fast breathing in ages 7–59 days.

§ Oral amoxicillin was unavailable

∞ Injection gentamicin was unavailable

Training to ASHA was given only at the time of recruitment and since there was no reorientation or supportive supervision, HBNC program was not executed efficiently. Essential equipment such as digital thermometer, weighing scale and digital respiratory rate counter were not functional. Some posts of ASHA were vacant. ANMs were not given oral amoxicillin,

**Table 6. Challenges faced during the implementation of the project.**

| Challenges | Steps taken | Outcome |
|---|---|---|
| • HBNC Training had been given to ASHA only at the time of appointment. | • Re-orientation of ASHAs on above aspects was a regular agenda during cluster meetings with success story sharing. | • Number of PSBI identification recorded an increase from 1.6% (of live births) in quarter 1 of establishment phase to 9.9% in quarter 1 of implementation phase. |
| • ASHAs were only weighing the baby and doing verbal inquiry from mothers about the health of the baby. They were not able to carry out all the recommended HBNC activities. | • Training and continuous re-orientation of ASHA with hands-on training in field and cluster meetings for: | • Number fast breathing cases identification increased from zero in quarter 1 establishment phase to 14.9% PSBI cases in quarter 1 of implementation phase duration. (**Table 3**) |
| | ➢ Counting respiratory rate | |
| | ➢ Correct method to record temperature and weight | |
| • HBNC recording form did not have space to record weight, temperature and respiratory rate. | • ASHA were requested to document the recordings on the HBNC forms. | |
| • ANMs, medical officers (excluding pediatricians) and staff nurses were "hesitant" to administer inj. gentamicin to infants due to fear of adverse events. | • Medical Superintendent motivated them for administration of inj. gentamicin through: | • Actual number of cases administered simplified treatment at SC/PHC/CHC increased to even though the proportion remained almost the same. (**Table 3**) |
| • ANMs also perceived that the community may not be willing for their newborns treated by them, especially administration of injection gentamicin. | ➢ Revision of existing government guidelines and role of ANM in simplified treatment of PSBI | • Pre-referral drug was administered to 2/8 cases of PSBI in establishment phase which increased to 14/25 cases in implementation phase (**Table 3**). Out of these, 02 pre-referral drugs were given by ANM. |
| | ➢ Addressing safety concerns over administration of gentamicin. | |
| | ➢ Sharing success stories of administration of simplified treatment and pre-referral dose by ANMs. | |
| | • Certificate of appreciation was awarded to ANMs who gave simplified treatment or administered pre-referral dose. | |
| | • A "**job aid**" was provided to ANMs and Medical Officers depicting guidelines to classify and manage a case of PSBI. | |
| • No provision for providing oral amoxicillin dispersible tablets and inj. gentamicin to ANMs even though it was available in government supply. One ml syringe was not available in government supply | • Oral amoxicillin dispersible tablets and injection gentamicin and 1 ml syringe with 26-gauge needle were provided through the research funds at the health facilities and ANMs in the establishment phase. | • Injection gentamicin was issued to ANMs by government pharmacists |
| | | • In Implementation phase government supplied oral amoxicillin dispersible tablets and injection gentamicin and 1 ml syringe with 26-gauge needle. |
| • Stock out of the HBNC visit recording forms hampered the documentation of the home visits by the ASHAs | • Government authorities were encouraged to regularize the supply of recording forms. | • HBNC forms were supplied to ASHA. |
| • Low awareness among mothers about neonatal danger signs. | • "**Mother card**" with information about signs of sepsis in newborns was developed and distributed by ASHAs to educate caregivers in village and mothers at the time of discharge after delivery from CHC/ PHC. | • Mother and family were able to identify signs of infection. Most families who identified sickness in their young infants did so in the implementation phase of the program. |
| • Low acceptance among the community for a referral to the government health facilities | | |
| • There was loss of critical time in transport between facilities, resulting in delay in treatment. | • The issue was discussed with State Government planners and policy makers to develop a mechanism to obtain information about the availability of a bed at the sick newborn care unit or referral hospitals | • Government endorsed simplified treatment to reduce the burden of cases at tertiary care facilities. |
| | | • Issue not completely resolved yet and the policy makers are in process of developing information system about availability of beds in NICU to cut short delay during transport |
| • ASHAs were not able to use the digital watch provided in the HBNC kit. On many occasions they were found non-functional. | • One to one training to each ASHA was given for counting of the respiratory rate by using their mobile phones. | Identification of cases of fast breathing cases by ASHA increased from establishment to implementation phase. |
| • Posts of ASHA were vacant in some villages. | • Responsibilities of HBNC visits in such areas were assigned to the ASHAs of the adjacent villages by health administrators. | • Number of first week visit after birth increased from 74.1% in quarter 1of establishment phase to 83.3% in quarter 1of implementation phase. |
| • There are some villages where ASHA are inactive. | | |

(*Continued*)

**Table 6.**  (Continued)

| Challenges | Steps taken | Outcome |
|---|---|---|
| • ASHAs had to participate in other government campaigns such as for immunization, non-communicable disease etc. | • During Cluster meetings, ASHA were reminded to continue the HBNC program follow-up. TSU staff was directly contacted by ASHAs telephonically when a sick infant was identified. | • There was continuous increase in 1st week visit by ASHA and PSBI case identification despite involvement in multiple programs. |
| | | • Supportive supervision and hand holding assisted ASHAs to develop multi-tasking abilities. |

Abbreviations:

ASHA Accredited Social Health activist

HBNC Home based Newborn Care

TSU Technical Support unit

injection gentamicin and 1 ml syringes for treat cases of PSBI. There was low awareness to mothers about signs of such illness.

## Discussion

In Lucknow public health setting, our data shows that it is feasible to implement the WHO guideline for PSBI management when referral is not feasible. First, the overall coverage rate in a birth cohort of 24448 was 53%, which is about half the expected, but we achieved a reasonably high coverage of 85% of PSBI cases in the implementation phase among those reached. Second, ASHAs were able to identify a large number of sick young infants under a strengthened HBNC. With training, supportive supervision and mentoring, the rate of identification of PSBI cases by ASHAs in the community increased from about 2% in the beginning to 8–10% at the end, which was similar to reported elsewhere [5]. Third, 95.2% (140/147) of young infants (age 7–59 days) with fast breathing were successfully and safely treated with oral amoxicillin at PHC/CHC level. Fourth, 94.2% (645/685) of young infants with signs of clinical severe infection whose parents refused referral advice to a hospital were successfully managed at an outpatient level with a combination of oral amoxicillin and intramuscular gentamicin with low case fatality rates. These high rates of treatment success and low case fatality are similar to data reported from other parts of Asia and Africa [13, 16, 17, 18, 19].

High acceptance of simplified treatment at the government facilities showed the confidence of caregivers in health care providers. This high acceptance is similar to that reported from both Asian and African sites, which have documented high refusal rates to acceptance of referral advice [13, 16, 17, 18, 19]. Several reasons for not accepting referral advice have been reported, such as distance to the hospital, cost of travel and treatment, concern about the quality of care or attitude of the health workers, lack of permission from family members, religious and cultural beliefs, and issues with lack of child care and other logistical problems [30, 31]. Unless these issues are addressed there will continue to be a large proportion refusing referral advice, especially in rural and less educated populations.

In the simplified treatment regimen, administration of daily injection gentamicin for 7 days was a perceived challenge and over 40% received at least up to 5 doses. Even though, compliance to all 7 doses was low, still very good treatment success of 92.4% was observed, supporting the WHO guideline that even two injections of gentamicin were effective [15].

The success rate of simplified treatment at PHC/CHC was higher and case fatality lower compared to those who were treated at a district hospital. This could be due to the fact that parents would opt for a higher facility for treatment if they perceived their infant to have more severe illness, or a delay caused by inappropriate care seeking and treatment elsewhere or

difficulty in getting transport to the hospital and sometimes not receiving pre-referral antibiotic treatment. But the most worrisome finding is high case fatality of those sick young infants whose parents refused formal treatment at all or sought private treatment, which could have been from inappropriate treatment provided. If the government is serious in addressing high neonatal mortality in the country then this needs to be addressed through community engagement strategies and improving quality of care in private sector. Empowering families through community activities in rural areas to create awareness to identify illness and seek appropriate care promptly is key to early management of sick young infants.

The main strength of our study was that this implementation research was executed within the existing government health system. Capacity strengthening of government functionaries and infra-structure improvement was carried out within a short period of nine months. Recognition of cases of PSBI by grass-root workers, ASHAs, almost tripled. Initial perception that giving injection gentamicin by the ANM in simplified treatment to young infants staying at home would be difficult proved incorrect. High coverage of simplified treatment was achieved by training, supportive supervision of physicians in the public health system who also acted as agents of change by setting an example. However, certain logistic weaknesses persisted despite close engagement of TSU with the health system. Most important was lack of medicines at the point of care. This resulted in use of alternate oral and injectable antibiotics. Simplified treatment was offered only when caregivers of cases of PSBI refused referral. Information about success rate of simplified treatment was not given to the parents. This resulted in caregivers still opting for treatment from private providers, who could have been unqualified or in the worst scenario not accepting any treatment. When simplified treatment of PSBI is rolled out at the state and national level, a strong communication component has to be integrated for community buy-in which is essential for successful reduction of neonatal mortality rate.

At the end of the implementation phase public health trainers also trained other doctors, nurses, ANMs and ASHAs in the remaining four blocks of Lucknow district that were not in the implementation phase. The government was requested to fast track procurement of oral amoxicillin and injection gentamicin. ANMs were issued gentamicin injection vials for use in cases of PSBI in the community.

During the establishment phase, only 1.6% cases of PSBI were identified as compared to 8.7% in the implementation phase, which was nearer to the PSBI rate of 7·6% (95% CI 6·1–9·2%) from a meta-analysis [4]. Thus prior to the implementation research, almost 3/4th cases of PSBI were missed. This would have contributed to high neonatal mortality rate in Uttar Pradesh. Improved detection of cases of PSBI in the implementation phase can be attributed to skills upgradation, supportive supervision and availability of complete HBNC kit of ASHA workers.

Our findings have shown that a substantial number of neonatal deaths can be averted through this simple intervention for management of PSBI cases. Our experience and that from other sites in India will hopefully give confidence to the government to seriously consider scaling up implementation of PSBI treatment guidelines through the country in areas where referral is not feasible. However, it will require continuous skills upgradation, supportive supervision and motivation of grass-root workers by their supervisors and physicians and ensuring continuous availability of supplies. A national and state level commitment is needed to reduce neonatal mortality through strengthening HBNC and effective implementation of simplified treatment of PSBI when referral is not feasible. Fortunately, the government of India already has issued a guideline [14, 20] but it needs more than just a document. Role of TSU in this implementation research was critical in achieving these positive results. When scaled up it is not essential to have a TSU, but technical support from experts will be needed to address the confidence issues in health workers and health system challenges to achieve high

coverage of treatment with high quality. The challenges faced during this study were addressed in a good working partnership between the TSU and the district managers and implementers.

We compiled 30 success stories of simplified treatment in local Hindi language. The state government obtained 1000 copies of these for distribution across the state to motivate health care workers. During the same time, government of India released national guidelines endorsing simplified treatment of PSBI [20]. The state government now has first-hand experience in rolling out simplified treatment under program setting in the pilot implementation area and with endorsement from central government, the state government is planning to expand simplified treatment for PSBI to remaining 74 districts of Uttar Pradesh.

## Conclusion

Simplified treatment for PSBI is feasible in public program settings in northern India with good cure rates. It required system strengthening and supportive supervision. There is a need to rollout this program on priority basis to reduce neonatal mortality.

## Supporting information

**S1 Table. The public health infrastructure available in Lucknow district and in 4 implementation blocks.**
(DOCX)

**S2 Table. Actions taken over responses of family of sick infant in community/ PHC/CHC.**
(DOCX)

**S3 Table. Initial training in four intervention blocks.**
(DOCX)

**S4 Table. Re-orientation training of ASHAs and ANMs in the intervention blocks (Nov 2017-Feb 2019).**
(DOCX)

## Acknowledgments

We thank the Ministry of Health and Family Welfare for giving us the permission and Late Dr. Anil Verma, General Manager, Child Health for support to conduct this implementation research. We also extend our gratitude to all the medical superintendents, medical officers, workers of the community process department, community health workers and parents of young infants of the rural blocks of Mall, Kakori, Gosainganj and Sarojini Nagar for their participation in the study.

## Author Contributions

**Conceptualization:** Shally Awasthi, Shamim Ahmad Qazi, Samira Aboubaker, Yasir Bin Nisar, Rajiv Bahl.

**Data curation:** Shally Awasthi, Naveen Kesarwani, Raj Kumar Verma, Luxmi Shanker Tewari, Ravi Krishna Mishra, Lalji Shukla, Arun Kumar Raut, Monika Agarwal.

**Formal analysis:** Shally Awasthi, Girdhar Gopal Agarwal.

**Funding acquisition:** Shally Awasthi.

**Project administration:** Shally Awasthi, Monika Agarwal.

**Resources:** Shally Awasthi.

**Supervision:** Shally Awasthi, Naveen Kesarwani, Raj Kumar Verma, Luxmi Shanker Tewari, Ravi Krishna Mishra, Lalji Shukla, Arun Kumar Raut.

**Validation:** Shally Awasthi, Monika Agarwal.

**Writing – original draft:** Shally Awasthi, Monika Agarwal.

**Writing – review & editing:** Shally Awasthi, Girdhar Gopal Agarwal, Shamim Ahmad Qazi, Samira Aboubaker, Yasir Bin Nisar, Rajiv Bahl, Monika Agarwal.

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
