## [Decision Letter · Decision Letter 0]

7 Apr 2020

PONE-D-20-05531

Identification and management of young infants with possible serious bacterial infection where referral was not feasible in rural Lucknow district of Uttar Pradesh, India: An implementation research.

PLOS ONE

Dear Dr. Awasthi,

Thank you for submitting your manuscript to PLOS ONE. After careful consideration, we feel that it has merit but does not fully meet PLOS ONE’s publication criteria as it currently stands. Therefore, we invite you to submit a revised version of the manuscript that addresses the points raised during the review process.

The reviewers have given very constructive and detailed comments. Kindly go through them and respond to them and revise the manuscript based on their comments. 

We would appreciate receiving your revised manuscript by May 10 2020 11:59PM. To enhance the reproducibility of your results, we recommend that if applicable you deposit your laboratory protocols in protocols.io, where a protocol can be assigned its own identifier (DOI) such that it can be cited independently in the future. For instructions see: http://journals.plos.org/plosone/s/submission-guidelines#loc-laboratory-protocols

We look forward to receiving your revised manuscript.

Kind regards,

Vijayaprasad Gopichandran

Academic Editor

PLOS ONE

Journal Requirements:

Reviewers' comments:

Reviewer's Responses to Questions

**Comments to the Author**

1. Is the manuscript technically sound, and do the data support the conclusions?

Reviewer #1: Yes

Reviewer #2: Yes

2. Has the statistical analysis been performed appropriately and rigorously? 

Reviewer #1: Yes

Reviewer #2: Yes

3. Have the authors made all data underlying the findings in their manuscript fully available?

Reviewer #1: No

Reviewer #2: Yes

4. Is the manuscript presented in an intelligible fashion and written in standard English?

Reviewer #1: Yes

Reviewer #2: Yes

5. Review Comments to the Author

Reviewer #1: The manuscript addresses an important implementation research undertaken in one of the challenging areas of India.

The following comments are to be addressed:

1. The manuscript primarily presents the quantitative findings from the study. The qualitative component of the study is not presented adequately.

2. The findings from Baseline, Midline survey have not been presented adequately. The findings of the interview and data collection from mothers have not been presented adequately.

3. The findings presented in the tables vary in terms of the differ in periods- somewhere the establishment phase and implementation phase have been combined, while at other places they are presented separately. A consistent pattern shall be useful.

4. Results section: The findings may start with the findings of the PSBI management and the variations or potential reasons may be explained from the qualitative and process observations.

5. The performance of the ANMs (10.5%) for administration of gentamicin should have been furthered explained with the trend across the implementation period. As this is important and relevant from other parts of India and other LMICs.

6. Overall the detection of PSBI cases was lower than anticipation. Possible explanations for this may have to be explored.

7. The variations across the 4 blocks and facilities, was there noticeable differences in the performance - detection of PSBI cases, management and compliance. If any variation observed, what were the possible reasons, as these learning shall inform the future scaling up.

8. Supplementary Table 2: The language need to be appropriate and consistent. The abbreviations need to be expanded below.

Other specific comments are in the text as comments/edits.

Reviewer #2: The paper provides outcomes of an operational research on identification and management of possible serious bacterial infection conducted in Uttar Pradesh, India in a community-based setting where referral is not possible. The study was well planned and executed and aligned with the government program for addressing infant mortality. The treatment protocol followed the WHO guidelines and guidelines issued by Ministry of Health in India. The article can be published in the journal after the authors make following revisions provided below.

Introduction

• Line 40: This line is not clear “mortality rate in India is 33 versus 41 per 1000 live births’ what is India’s IMR, UP IMR, etc.

• The introduction talks about the HBNC program, how is it done, ASHA, etc. but it is not clear from the write-up why do they want to do this study. Are there studies already available or no data or what are the lacunae in the existing evidence? This is totally missing: the rationale of doing this study?

• The authors seem confused whether it is operational research or implementation research.

• Line 69-71: the authors suddenly jumped from operational research to implementation research. If the authors want to describe about the implementation, they need to write about it separately (a separate para in the introduction section) because the operational study was done in UP and IR was done in other states.

Methodology

• Line 76: This is not clear. Whose birth rate are we talking about? Uttar Pradesh or Lucknow‘ has a crude birth rate of Uttar Pradesh is 25.9 per 1000 population’.

• Line 116: the reference has been wrongly put. ‘at SC. [24]They’

• Line 82-84: the program and its activities have been detailed excessively. This is too much for the reader to absorb. Can the additional details of the program be explained separately or in the introduction. Because in the methodology section, it’s the methodology of the study that should come and not the program.

• What mechanisms were followed by ASHAs for not missing out any case for investigation, as case identifications were done at the community level.

• Apart from training and ongoing orientation, how it was ensured that investigation conducted by ASHAs in the community were accurate and there were no ‘false negative’ cases? Also, how many cases identified by ASHAs were ‘false positive’?

• What strategies were adopted to improve treatment compliance and motivating parents to come for regular treatment of their child as only around 54% of children took all 14 doses of oral amoxicillin and only 16.3% of children received all 7 doses of Gentamycin injection?

Results

• Can table 3 be reduced or broken down into 2 different tables? It is too long and difficult to read.

• It would be good if comparative findings of two cross-sectional surveys-baseline and midline are presented to present change in the knowledge and skills of ASHAs and ANMs in terms of identification and management of PSBI as this is important before transition into the implementation phase.

• Reasons need to be provided for infants (6.4%) who did not complete the mandatory follow ups on Day 4 and Day 8.

• Also, reasons need to be explained for less compliance.

Discussion

• To assist policy decision, as a future research, authors can consider conducting the ‘cost effectiveness analysis’ of identification and management of young infants with PSBI, where referral was not feasible.

6. PLOS authors have the option to publish the peer review history of their article (what does this mean?). If published, this will include your full peer review and any attached files.

Reviewer #1: No

Reviewer #2: Yes: Dr. Sunil Mehra, Dr. Rajesh Kumar Sinha, Dr. Shantanu Sharma, Dr. Vaibhav Rastogi

---

## [Author Response · Author response to Decision Letter 0]

20 May 2020

The response to reviewers comments have been addressed and are being submitted as a separate file

---

## [Editor Report · Decision Letter 1]

21 May 2020

Identification and management of young infants with possible serious bacterial infection where referral was not feasible in rural Lucknow district of Uttar Pradesh, India: An implementation research.

PONE-D-20-05531R1

Dear Dr. Awasthi,

We are pleased to inform you that your manuscript has been judged scientifically suitable for publication and will be formally accepted for publication once it complies with all outstanding technical requirements.

With kind regards,

Vijayaprasad Gopichandran

Academic Editor

PLOS ONE
---

## [Editor Report · Acceptance letter]

26 May 2020

PONE-D-20-05531R1 

Identification and management of young infants with possible serious bacterial infection where referral was not feasible in rural Lucknow district of Uttar Pradesh, India: An implementation research. 

Dear Dr. Awasthi:

I am pleased to inform you that your manuscript has been deemed suitable for publication in PLOS ONE. Congratulations! Your manuscript is now with our production department. 

With kind regards,

on behalf of

Dr. Vijayaprasad Gopichandran 

Academic Editor

PLOS ONE